# Carbon Nanodots-Embedded Pullulan Nanofibers for Sulfathiazole Removal from Wastewater Streams

**DOI:** 10.3390/membranes12020228

**Published:** 2022-02-16

**Authors:** Muhammad Omer Aijaz, Munir Ahmad, Mohammad I. Al-Wabel, Mohammad Rezaul Karim, Adel R. A. Usman, Abdulaziz K. Assaifan

**Affiliations:** 1Center of Excellence for Research in Engineering Materials (CEREM), Deanship of Scientific Research (DSR), King Saud University, P.O. Box 800, Riyadh 11421, Saudi Arabia; maijaz@ksu.edu.sa; 2Soil Sciences Department, College of Food & Agricultural Sciences, King Saud University, P.O. Box 2460, Riyadh 11451, Saudi Arabia; amunir@ksu.edu.sa (M.A.); malwabel@ksu.edu.sa (M.I.A.-W.); adosman@ksu.edu.sa (A.R.A.U.); 3K.A. CARE Energy Research and Innovation Center, Riyadh 11451, Saudi Arabia; 4Department of Soils and Water, Faculty of Agriculture, Assiut University, Assiut 71526, Egypt; 5King Abdullah Institute for Nanotechnology, King Saud University, P.O. Box 2455, Riyadh 11451, Saudi Arabia; aassaifan@ksu.edu.sa

**Keywords:** composite nanofibers, nanodots, polysaccharide pullulan, sulfathiazole antibiotics

## Abstract

Carbon nanodots (CNDs)-embedded pullulan (PUL) nanofibers were developed and successfully applied for sulfathiazole (STZ) removal from wastewater streams for the first time. The CNDs were incorporated into PUL at 0.0%, 1.0%, 2.0%, and 3.0% (*w*/*w*) to produce M1, M2, M3, and M4 nanofibers (PUL-NFs), respectively. The produced PUL-NFs were characterized by scanning electron microscopy (SEM), transmission electron microscopy (TEM), Fourier transform infrared spectroscopy (FTIR), X-ray diffraction analysis (XRD), thermal gravimetric analysis (TGA) and Differential scanning calorimetry (DSC) and applied for STZ removal from aqueous solutions through pH, kinetics, and equilibrium batch sorption trials. A pH range of 4.0–6.0 was observed to be optimal for maximum STZ removal. Pseudo-second order, intraparticle diffusion, and Elovich models were suitably fitted to kinetics adsorption data (*R*^2^ = 0.82–0.99), whereas Dubinin–Radushkevich, Freundlich, and Langmuir isotherms were fitted to equilibrium adsorption data (*R*^2^
*=* 0.88–0.99). STZ adsorption capacity of PUL-NFs improved as the amount of embedded CNDs increased. Maximum STZ adsorption capacities of the synthesized PUL-NFs were in the order of: M4 > M3 > M2 > M1 (133.68, 124.27, 93.09, and 35.04 mg g^−1^, respectively). Lewis acid–base reaction and π-π electron donor–acceptor interactions were the key STZ removal mechanisms under an acidic environment, whereas H-bonding and diffusion were key under a basic environment. Therefore, CNDs-embedded PUL-NFs could be employed as an environmentally friendly, efficient, and non-toxic adsorbent to remove STZ from wastewater streams.

## 1. Introduction

The pharmaceutical industry has developed over 250 essential antibiotics for better public health and life quality with an annual consumption of more than 63 tons [1,2]. However, 30% of antibiotics are metabolized and absorbed in the targeted organism of the body and the remaining active compounds are excreted as microcontaminants in soil and water ecosystems. This huge amount of excreted antibiotics, released mainly from the wastewater discharge of hospitals, veterinary farms, and municipal effluents [3], are introduced into surface and drinking water [4,5]. Sulfathiazole (STZ) is one of the commonly used topical antimicrobials, with short-acting sulfonamide antibiotics, generally used in vaginal and urinary tract infections. The structure of STZ (C_9_H_9_N_3_O_2_S_2_) contains a heterocyclic (1,3-thiazole) compound with a 4-aminobenzenesulfonamido group (Figure 1). Exposure to high levels of STZ is harmful to human health and ecosystems. Hence, STZ removal from wastewater streams, prior to their disposal into water bodies, is of critical importance.

Various methods have been used by different researchers for antibiotics removal, such as biodegradation, electrochemical oxidation, photocatalysis, ion exchange, reverse osmosis, adsorption, and membrane processes [6,7,8,9,10]. However, these methods have some disadvantages when it comes to effective and low-cost removal of antibiotics from wastewater treatment plants. Nanofibers (NFs)-based membranes have recently emerged as efficient adsorbents for environmental contaminants, such as metal ions, dyes, and antibiotics [11,12,13,14,15,16]. Removal of antibiotics using electrospun NFs is still a new and economical method that requires further exploration due to the versatile range of NFs properties, such as porosity, surface area, narrow diameter, permeability of gas, and smaller fiber-to-fiber pore size. NFs can easily be altered by modifying the fiber size, orientation, and surface functionality, and by manipulating the composition of the different required materials [17]. Hitherto, limited research work has been available in the literature in which NFs have been used as antibiotic removal agents. For example, electrospun NFs incorporated with iron oxide were prepared by electrospun-solvothermal methods and checked for tetracycline antibiotics removal capacities from aqueous solution, and the maximum adsorption capacity calculated from Langmuir isotherm model was 315.31 mg g^−1^ [18]. Another study dealt with activated carbon loaded with electrospun NFs membranes to remove fluoroquinolones antibiotics from wastewater, and adsorption study outcomes were in the range of 93–112 mg g^−1^ [19]. Recently, electrospun spongy zero-valent iron mats were prepared and applied for the removal of STZ by a combination of adsorption and electro-catalytic oxidation [14], and the adsorption capacity was found to be around 35 mg g^−1^.

Pullulan (PUL) is an extracellular biodegradable linear polysaccharide produced by an organism called *Aureobasidium pullulans*. The PUL structure consists of maltotriose repeating units (*α*-(1,4)) linked alternatively by an *α*-(1,6) glycosidic bond [20,21,22] (Figure 1). The linear alternative linkage pattern provides PUL with good solubility, processability and spinnability. Pullulan is a tasteless, white-colored powder having characteristic properties of thermal stability (250–280 °C), binding and adhesion, biocompatibility, and non-toxicity with good processability to form fiber and films. PUL is a material of choice for a wide range of applications, including as an additive for cosmetics, food ingredients, oxidation-prevention agent in medicines, coatings, packaging materials for food and drugs, fertilizer binder, contact lenses, biodegradable foil, plywood, biomedical applications, and so on.

Carbon nanodots (CNDs) are nanosized carbon nanoparticle materials with sizes in the range of 1 nm to 60 nm, averaging 10 nm [23]. CNDs have exceptional properties, including biocompatibility, chemical inertness, non-toxicity, ease of water dispersibility, high surface area, and low cost. CNDs have the capability to enhance the removal and separation capacity of pollutants from environments, due to ultra-high surface passivation with high specific area. However, carrier mediums, such as NFs, are often required to hold CNDs to avoid aggregation. Therefore, the incorporation of CNDs within NFs may improve the adsorption efficiency of NFs for STZ through the combined properties of NFs and CNDs. Recently, in a few studies, the incorporation of CNDs in electrospinning NFs found their application in different fields. For example, Zhai et al. [24] successfully prepared the hybrid NFs-CNDs and polyvinylpyrrolidone through electrospinning to study the optical properties. Recently, Nie et al. [25] showed that embedding CNDs into electrospun polyacrylonitrile (PAN)-NFs resulted in improved inactivation of antibacterial agents. Similarly, a few more studies have demonstrated that CNDs can successfully be combined with NFs membranes based on polyvinyl alcohol, PAN, and a blend of polyacrylic acid/PAN with improved physical and structural properties [26,27,28]. However, CNDs-embedded NFs have not yet been used for the adsorption of STZ antibiotics from water. Therefore, the CNDs were prepared and embedded into electrospun NFs membranes to fabricate PUL-NFs composite adsorbents. The synthesized PUL-NFs were tested to explore the chemical, structural, and physical composition and applied for the removal of STZ from aqueous solution through pH, isotherm, and kinetic batch adsorption trials. Further, the adsorption data were simulated by using various isotherm and kinetics models to understand the removal of STZ by PUL-NFs.

## 2. Materials and Methods

### 2.1. Chemicals 

Acetic acid (CH_3_COOH), ethanol (C_2_H_5_OH), methanol (CH_3_OH), citric acid (C_6_H_8_O_7_), and sodium hydroxide (NaOH) were purchased from Thermo Fisher Scientific Company, Waltham, MA, USA. STZ (C_9_H_9_N_3_O_2_S_2_) and reagent grade chitosan (C_6_H_11_NO_4_)_n_ with low molecular weight were obtained from Sigma Aldrich, Merck KGaA, Burlington, Middlesex County, MA, USA. Pullulan (PF-20 grade) (C_6_H_12_O_5_)_n_ was purchased from Hangzhou Focus Corporation (Zhejiang, China). Electrospun PUL-NFs were produced by using an electrospinning machine (MECC, Fukuoka, Japan; model NF-500). In adsorption experiments, purified deionized water (milliQ, Merck, Darmstadt, Germany) was used.

### 2.2. Synthesis of CNDs

For the synthesis of CNDs, wastage from date palm trees was collected; leaves were separated, dried, crushed, and segregated through a sieve of 0.6 mm mesh size. The sieved date palm leaf waste was heated at 600 °C for 180 min in an airtight container by using a high-temperature tube furnace. Upon cooling, the produced carbonaceous material (biochar) was washed with deionized water and dried in an oven at 45 °C. Hereafter, the material was refined at 700 rotations per minute for 30 min using a ball-mill (PULVERISETTE-7, Fritsch, Weimar, Germany) and used for the synthesis of CNDs. The CNDs were synthesized by adapting the procedure reported by Li et al. [29]. Specifically, citric acid (2.0 g) and the refined carbonaceous material (5.0 g) were mixed in 100 mL of deionized water, and the suspension was stirred for 3 h. Then, 1.0 g of chitosan dissolved in 2.0% acetic acid was poured into the suspension under continuous stirring. After another 30 min of stirring, 100 mL of 1.2% NaOH solution was added to the suspension. The suspension was mixed thoroughly and heated in an airtight container at 180 °C for 2.0 h. The fabricated CNDs were separated, washed with methanol thrice, followed by deionized water, and stored in airtight containers.

### 2.3. Synthesis of CNDs-Embedded Pullulan NFs

To fabricate CNDs-embedded NFs, PUL was dissolved in deionized water to prepare a 14% PUL solution (*w*/*v* ratio) by stirring on a magnetic stirrer at 23 ± 2 °C. Thereafter, the synthesized CNDs were suspended into the PUL solution at the rates of 0.0%, 1.0%, 2.0%, and 3.0% (*w*/*v* ratio) using a mechanical stirrer. The homogenized PUL-CNDs suspension was filled in a 5.0 mL syringe with a needle diameter of 0.6 mm for electrospinning and PUL-NFs were fabricated. The electrospinning parameters were selected as 25 kV of applied voltage, 0.8 mL h^−1^ of flow-rate, and 15 cm of collector-to- needle distance [22]. The synthesized PUL-NFs were named as M1 (0.0% CNDs), M2 (1.0% CNDs), M3 (2.0% CNDs), and M4 (3.0% CNDs). The stepwise synthesis process can be seen in Figure 1. Finally, the electrospun PUL-NFs were detached from the flat collector and stored for further characterizations.

### 2.4. Characterization

#### 2.4.1. FE-SEM and TEM

The morphology of the prepared PUL-NFs was studied using a field-emission scanning electron microscope (FE-SEM: JSM-7600, JEOL, Tokyo, Japan). Small pieces of the prepared PUL-NFs were fixed onto a sample holder with the help of carbon tape. Before inserting the holders into the FE-SEM chamber, samples were coated with platinum to increase electrical conductivity of the sample during the analysis. Afterward, coated samples were analyzed by FE-SEM under a high vacuum. Furthermore, detailed morphological observations were carried out using Transmission Electron Microscopy (TEM: JSM-2100F, JEOL, Tokyo, Japan). TEM samples were prepared by directly electrospinning a few PUL-NFs onto a copper grid placed on top of the electrospinning collector. A few PUL-NFs were achieved on the grid by electrospinning for a short period of time.

#### 2.4.2. XRD, FTIR, and TGA

To identify the functional groups of prepared PUL-NFs, the Fourier transformed infrared (FT-IR: VERTEX-70, Bruker, Billerica, MA, USA) was utilized in the range of 600–4000 cm^−1^. Thermal behavior of prepared PUL-NFs was studied in a temperature range of 25 to 600 °C by using a thermal gravimetric analysis machine (TGA: Q600, TA Ins., New Castle, DE, USA) at 20 °C per min. A small amount of electrospun PUL-NFs was placed in a ceramic pan and the test was run under an inert (N_2_) atmosphere. The structure and crystallinity of the PUL-NFs were investigated using an X-ray diffractometer (XRD: XRD-7000, Shimadzu, Kyoto, Japan). The samples were analyzed in the 2.0–80 2θ-degree range with continuous scanning at the rate of 2°/min.

### 2.5. Adsorption Batch Studies

The 1000 mg·L^−1^ stock solution of STZ was prepared in 1.0% CH_3_OH. The working standards of 0–150 mg·L^−1^ of STZ were prepared by diluting the stock solution in deionized water and adjusted pH as needed. 15 mg of the PUL-NFs adsorbent was added to 20 mL of STZ solution in a plastic tube. The solution containing the adsorbent was shaken for a certain interval of time at 23 ± 2 °C. Thereafter, the adsorbent was removed from the solution through 450 nm syringe filters and stored in amber color bottles at 4 °C. The remaining amount of STZ inside the solution was studied by using a high-performance liquid chromatograph (HPLC: LC-2030C, Shimadzu, Kyoto, Japan). The HPLC instrument was calibrated by running the STZ standards, and a calibration curve of *R*^2^ ~ 0.99 was acquired.

The impacts of solution pH on STZ sorption were investigated in batch studies using an initial pH range of 4.0–10 with an initial STZ concentration of 100 mg·L^−1^. The dynamics of STZ sorption were studied in kinetic sorption batches. An initial STZ concentration of 100 mg·L^−1^ and pH of 4.5 were used in kinetic batch studies. The samples were withdrawn after 0.0, 5.0, 10, 30, 60, 120, and 180 min. The equilibrium sorption batches were conducted with initial STZ concentrations of 0.0, 1.0, 5.0, 20, 50, 100, and 150 mg·L^−1^ with an initial solution pH of 4.5. All the sorption batch trials were conducted in three replications, including a blank (without adsorbent), at 23 ± 2 °C to confirm repeatability. The amount of STZ adsorbed onto the PUL-NFs was calculated by using Equation (1) [30]:(1)$$q_{e}=\left[\frac{C_{o}-C_{e}}{m}\right]\times v$$
where *C_o_* is initial and *C_e_* is equilibrium STZ concentration (mg·L^−1^), *m* is the mass of the adsorbent used in sorption trials (g), and *v* is the volume of STZ solution used (L).

### 2.6. Kinetic Modelling

The kinetics sorption data were simulated with mathematical models to explore the STZ sorption dynamics as follows [31]:(2)$$\mathrm{First}\;\mathrm{order}\;\;\;\;lnq_{t}=lnq_{o}-k_{1}t$$
(3)$$\mathrm{Sec}\mathrm{ond}\;\mathrm{order}\;\;\;\;\frac{1}{q_{t}}=\frac{1}{q_{o}}-k_{2}t$$
(4)$$\mathrm{Pseudo}-\mathrm{first}\;\mathrm{order}\;\;\;\;ln\left(q_{e}-q_{t}\right)=lnq_{e}-k_{1}^{’}t$$
(5)$$\mathrm{Pseudo}-\mathrm{second}\;\mathrm{order}\;\;\;\;\frac{t}{q_{t}}=\frac{1}{k_{2}^{’}q_{e}^{2}}+\frac{1}{q_{e}}t$$
(6)$$\mathrm{Elovich}\;\;\;\;q_{t}=\frac{1}{\beta }ln\;\left(\alpha \beta \right)+\frac{1}{\beta }\;lnt$$
(7)$$\mathrm{Power}\;\mathrm{function}\;\;\;\;lnq_{t}=lnb+k_{f}\left(lnt\right)$$
(8)$$\mathrm{Intraparticle}\;\mathrm{diffusion}\;\;\;\;q_{t}=c+k_{id}t^{0.5}$$
where *t* is time, $q_{t}$ is the STZ amounts adsorbed at time *t* (mg·L^−1^), $q_{o}$ is amount of STZ adsorbed at time zero (mg·L^−1^), $q_{e}$ is the maximum sorption capacity (mg·g^−1^), $k_{1}$ is the rate constant of first order, $k_{2}$ is the rate constant of second order, $k_{1}^{’}$ is rate constant of pseudo-first order, $k_{2}^{’}$ is rate constant of pseudo-second order, *α* is rate of initial sorption (mg·g^−1^ min^−1^), *β* is sorption constant, $k_{f}$ is rate coefficient (mg·g^−1^·min^−1^), *b* is rate constant, *c* is constant, and $k_{id}$ is rate constant of intraparticle diffusion ([mg·g^−1^]^−0.5^).

Equations (9) and (10) were employed to calculate standard error of estimate (*SEE*) and coefficient of determination (*R*^2^), respectively, as indicated below:(9)$$SEE=\sum_{i=1}^{n}{\left(q_{em}-q_{ec}\right)}^{2}$$
(10)$$R^{2}=\frac{{\left(q_{em}-\bar{q_{ec}}\right)}^{2}}{\sum^{\;}{\left(q_{em}-\bar{q_{ec}}\right)}^{2}+{\left(q_{em}-q_{ec}\right)}^{2}}$$
where *n* is for number of measurements, $q_{em}$ is actual sorption capacity (mg·g^−1^), and $q_{ec}$ is calculated sorption capacity (mg·g^−1^).

### 2.7. Isotherm Modelling

The equilibrium sorption data were simulated by using nonlinear forms of various isotherm models, as shown below [31]:(11)$$\mathrm{Langmuir}\;\;\;\;q_{e}=\frac{Q_{L}C_{e}K_{L}}{1+K_{L}C_{e}}$$
(12)$$\mathrm{Freundlich}\;\;\;\;q_{e}=K_{F}C_{e}^{1/n}$$
(13)$$\mathrm{Temkin}\;\;\;\;q_{e}=\frac{RT}{b}lnAC_{e}$$
(14)$$\mathrm{Dubinin}–\mathrm{Radushkevich}\;\;\;\;q_{e}=Q_{D}\exp(-B_{D}\left[RTln\;\left(1+\frac{1}{C_{e}}\right){]}^{2}\right)$$
where $Q_{L}$ and $Q_{D}$ are highest sorption amounts (mg·g^−1^), $K_{L}$ is sorption constant at equilibrium (L·mg^−1^), 1/*n* is the linear component of Freundlich, $K_{F}$ is sorptive affinity of Freundlich (L·g^−1^), *R* is universal gas constant, *A* is binding constant (L·mg^−1^), *b* is adsorption heat (J·mol^−1^), *T* is absolute temperature, and *B_D_* is mean free energy of adsorption (kJ·mol^−1^).

The binding energy (*E*) and separation factor (*R_L_*) were calculated, as shown below:(15)$$E=\frac{1}{\sqrt{2B_{D}}}$$

## 3. Results and Discussion

### 3.1. Characterization

#### 3.1.1. SEM and TEM Analyses

The morphological images of PUL-NFs M1, M2, M3, and M4 are provided in Figure 2a–d. A smooth and fine structure of fibers was observed with an average diameter of ~140 nm in M1 (Figure 2a), while the addition of CNDs showed an increasing trend of roughness as well as fiber diameter in M2, M3, and M4 (Figure 2b–d). Red circular marks in Figure 2b–d clearly indicated the presence of CNDs, as more particle agglomerations were found with increasing the percentile of CNDs. Therefore, to avoid further agglomeration of CNDs in the PUL NFs, no more (>3%) CNDs were added in the PUL solution. The diameters of NFs containing the particles were in the range of ~145, 178, and 195 nm for M2, M3 and M4, respectively.

Figure 3a–c shows the in-depth morphological images of PUL-NFs to further confirm the presence of CNDs in PUL-NFs at 50, 20 and 10 nm of magnification. In Figure 3a–c, red circles indicate the incubated CNDs with single NFs.

#### 3.1.2. XRD, FTIR, and TGA Analyses

The crystallinity and structural composition of the fabricated PUL-NFs as studied by XRD are presented in Figure 4a. The NFs without CNDs displayed an amorphous structure. The XRD patterns of materials M2, M3, and M4 exhibited two very intense peaks at 44.5° and 78.1°, depicting the presence of CNDs. These peaks were owing to the occurrence of crystalline graphite-like carbon nanofibers. Similarly, a peak appearing at 65.2° in all the PUL-NFs samples was attributed to crystalline carbon (turbostratic carbon). Interestingly, the intensity of the peaks representing the crystalline graphite-like carbon nanofibers and turbostratic carbon increased with increasing the amount of embedded CNDs in the PUL-NFs. The diffractions peaks appearing at 34.8°, 37.1°, 39.5°, 58.8°, and 68.9° were assigned to graphene quantum dots. Therefore, these results confirmed the presence of CNDs in the fabricated PUL-NFs.

The ATR-FTIR curves of the prepared PUL-NFs are displayed in Figure 5. The identical spectral peaks for PUL components were observed in all the curves from 600 to 1500 cm^−1^. The peaks at 800, 755, and 932 cm^−1^ were attributed to *α*-glucopiranosid units, *α*-1,4, and *α*-1,6 glucosidic bonds, respectively. Similar spectral features were observed for M1, M2, M3, and M4 as well, due to the fact that PUL and CNDS consist of carbon, oxygen, and hydrogen atoms. Broader peaks of hydroxyl bond and C–O stretching were observed in the range of 3000–3600 cm^−1^ and 1000–1260 cm^−1^, respectively. C–H/C–H_2_ stretching vibration peaks present at 2850–3000 cm^−1^ and 1300–1500 cm^−1^ ranges are attributed to stretching vibration and bond deformation of C–H and C–H_2_ groups [22]. Broad peaks of the hydroxyl group at 3360 cm^−1^ and C–O stretching at 1020 cm^−1^ of PUL were reduced gradually as the amount of CNDs increased. The reduction in peaks might be due to the addition of CNDs to the PUL polymer.

TGA, DTA, and DSC curves for M1, M2, M3, and M4 composite PUL-NFs under inert N_2_ gas from 25 to 600 °C are presented in Figure 6. Observations showed that degradation started from the removal of moisture content in the range of 50–100 °C. However, as the temperature increased, the onset decomposition curve started to be observed at 288.1, 290.8, 290.2, and 289.3 °C for M1, M2, M3, and M4, respectively. Complete weight loss was observed at a temperature above 300 °C in all the PUL-NFs. TGA and DTA observations are presented in Table 1, indicating that as the percentage of CNDs increased, the onset temperature (T_onset_) (Figure 6A) and maximum peak degradation temperature (T_peak_) (Figure 6B) were increased along with the residue percentage, due to higher amounts of CDQs. However, M4 showed the minimum T_onset_ and T_peak_ compared to M2 and M3, which could be attributed to the loose hydrogen bonding between the fibers due to aggregation.

In the DSC curves (Figure 6C), slight increases in glass transition (T_g_) and melting temperature (T_m_) were observed as the percentage of CNDs increased, and their details are presented in Table 1. The improvement in thermal stability could be attributed to more chain interaction between PUL and CNDs in composite PUL-NFs membranes.

### 3.2. STZ Adsorption Experiments

#### 3.2.1. Effect of pH on STZ Adsorption

Akin to charge density, binding sites, and aqueous chemistry, solution pH is of critical importance in removing the sulfa drugs from water. Therefore, the impact of the initial solution pH was studied in the range of 4.0–10 (Figure 7). Overall, the highest STZ adsorption using all the PUL-NF types was noted in the pH range of 4.0–6.0. The adsorption of STZ onto each of the PUL-NF types decreased slowly as the pH value increased from 4.0 to 6.0, while it dropped sharply as the solution pH increased further (6.0 to 10). The sudden decline in STZ adsorption above pH 6.0 was explained through the point of zero charge (pzc) of the adsorbents [2]. The pH at pzc for M1, M2, M3, and M4 was found as 6.58, 6.83, 6.91, and 7.16, respectively. It has been reported earlier that a solution pH below pzc yields a positively charged surface of the adsorbents and vice versa [32]. Therefore, the predominance of negative charges on the surface of the synthesized PUL-NFs is evident above pH 6.0, as explained by pzc. On the other hand, the speciation of STZ is mainly pH dependent of the solution, owing to the amphoteric nature of STZ molecules. It has been reported that the STZ molecule exits as a neutral ion (STZ^0^) in the pH range of 3.0–6.0; however, it transforms to cation (STZ^+^) or anion (STZ^–^) when pH decreases below or increases above this range, respectively, due to dissociation/protonation [33]. Thus, a sudden decrease in STZ adsorption above pH 6.0 was due to electrostatic repulsion between STZ^–^ molecules and negatively charged adsorbents. However, the higher adsorption of STZ in acidic pH could be owing to the electron donor–acceptor (EDA) π-π interlinkages, because the STZ molecules act as strong π-acceptor at lower pH levels due to the presence of protonated amino group (−NH^3+^), whereas CNDs serve as π-donor [34,35]. Further, the STZ adsorption capacity of the PUL-NFs improved with increasing the amount of CNDs. The STZ adsorption capacity of M1 at pH 4.0 was recorded as 26.53 mg g^−1^, which increased to 62.70, 87.96, and 99.59 mg g^−1^ in M2, M3, and M4, respectively. Overall, a 3% addition of CNDs (M4) adsorbed fourfold higher STZ from aqueous solution than pristine PUL-NFs (M1). The improved STZ adsorption efficiency of PUL-NFs with CNDs enrichment could be owing to enhanced surface area, higher π-donation capacity, and increased surface functional groups.

#### 3.2.2. Adsorption Kinetics

The dynamics of STZ adsorption onto the fabricated PUL-NFs were studied in kinetic batch adsorption trials at a constant temperature and solution pH. Three distinct adsorption phases were observed in STZ adsorption dynamics for the adsorbents M3 and M4 (Figure 8a). A rapid adsorption phase was seen from 0.0–10 min, followed by a slow adsorption phase (10–60 min), and an equilibrium phase (60–180 min). The rapid adsorption phase in M3 and M4 during the initial 10 min could be due to the abundance of the sorption sites, which were occupied by STZ molecules over time, finally reaching the equilibrium stage. Unlike M3 and M2, the rapid adsorption phase was missing for the adsorbents M1 and M2, where adsorption slowly increased from 0.0–120 min and attained equilibrium phase from 120–180 min. Overall, M4 exhibited the highest adsorption rate and adsorption capacity compared with the other tested adsorbents. The STZ adsorption data were simulated with kinetic models to further understand the adsorption process. The *SEE* values for all the kinetics models were <1.0, indicating lower chances of errors (Table 2).

The STZ adsorption data were well explained by pseudo-second order, intraparticle diffusion, and Elovich models, as depicted from *R*^2^ values (0.82–0.99) (Figure 8). First-order, power-function, second-order, and pseudo-first order models were unsuitable for describing the STZ adsorption onto the synthesized adsorbents. Thus, the best fitness of the pseudo-second order and Elovich model to STZ adsorption data suggested chemical adsorption, whereas the fitness of the intraparticle diffusion model suggested the involvement of the diffusion process in the STZ removal. The parameters obtained from the kinetics sorption modelling are shown in Table 3.

The pseudo-second order predicted *q_e_* was highest for M4 (98.58 mg·g^−1^), followed by M3 (89.94 mg·g^−1^), M2 (55.82 mg·g^−1^), and M1 (38.79 mg·g^−1^). Likewise, rate constants as expected by the pseudo-second order and Elovich models were the highest for M4 (*k*^2^ = 1.8 × 10^−^^3^ and *β* = 10.30, respectively). Akin to this, the diffusion constant (*c*) and initial sorption constants (*α*) representing the rate of initial sorption were also the highest for M4 adsorbent (26.94 and 18.09 mg·g^−1^ min^−1^, respectively). Thus, the highest rate constants and diffusion constants of M4 indicated its higher adsorption affinity for STZ compared with other adsorbents. The highest value of the intraparticle diffusion rate constant (*k^id^*) was observed in M4 (6.29 [mg·g^−1^]^−0.5^), followed by M3 (6.05 [mg·g^−1^]^−0.5^), M2 (5.19 [mg·g^−1^]^−0.5^), and M1 (2.35 [mg·g^−1^]^−0.5^), indicating a faster diffusion of STZ into the matrix of M4. Hence, kinetics modelling simulation suggested chemisorption and diffusion as the STZ removal mechanism from aqueous media using PUL-NFs.

#### 3.2.3. Equilibrium Adsorption

The isotherm adsorption studies were conducted with varying the initial STZ concentrations at constant pH, adsorption dose, and temperature. The STZ adsorption data were fitted to the nonlinear forms of Langmuir, Dubinin–Radushkevich, Freundlich, Temkin, and Redlich–Peterson models, and the resultant isotherms are exhibited in Figure 9. Generally, the amount of adsorbed STZ increased with increasing the initial STZ concentration [36]. A high affinity (H-type) isotherm was generated at lower initial STZ concentrations in M2, M3, and M4, suggesting the presence of an abundant adsorption site. However, as the initial STZ concentration increased beyond 10 mg·L^−1^, the shape of the isotherm transformed to L-type, which could be described by the lower availability of the adsorption sites. Nonetheless, the isotherm shape of M1 differed from M2, M3, and M4, and generated an L-type isotherm at the beginning, followed by an equilibrium phase. The missing H-type isotherm in M1 depicted its significantly lower STZ adsorption capacity than M2, M3, and M4. Nonlinear parameters after simulating STZ adsorption with isotherm models are given in Table 4.

Langmuir, Freundlich, and Dubinin–Radushkevich models were fitted well to the STZ adsorption data that was confirmed by the *R*^2^ values generated between the range of 0.88–0.99, whereas the Temkin model was marginally fitted to the adsorption data (*R^2^* = 0.77–0.88). The adsorbent M4 adsorbed almost fourfold higher STZ from aqueous solution than the pristine PUL-NFs. Overall, the maximum STZ adsorption capacities of the synthesized PUL-NFs were in the order of: M4 > M3 > M2 > M1, as suggested by the Langmuir model, which predicted *Q_L_* (133.68, 124.27, 93.09, and 35.04 mg·g^−1^, respectively), and the Dubinin-Radushkevich model, which predicted *Q_D_* (106.45, 97.05, 69.12, and 29.20 mg·g^−1^, respectively). A similar trend has also been observed in adsorption kinetics studies, where the addition of CNDs significantly increased the STZ adsorption capacity of PUL-NFs. Likewise, the Freundlich model, which predicted *K_F_*, demonstrated a similar trend, generating the highest value for M4 (26.48 L·g^−1^), followed by M3 (11.77 L·g^−1^), M2 (8.03 L·g^−1^), and M1 (4.06 L·g^−1^). Therefore, the fitness of the Langmuir and Freundlich isotherm models suggested both monolayer and multilayer adsorption of STZ onto the surface of used PUL-NFs. On the other hand, the lower values of the Dubinin–Radushkevich model, which predicted *E* (<8.0 kJ·g^−1^), suggested that the ion-exchange process was not involved in STZ removal [31]. To further explore the favorability of STZ adsorption by the prepared PUL-NFs, the Freundlich model, which predicted 1/*n*, was used. The estimated values for all the PUL-NFs adsorbents were <1.0 (0.35–0.49), suggesting higher sorption affinity of PUL-NFs towards STZ. Therefore, the results of equilibrium sorption batch studies revealed that STZ removal followed both single and multilayer adsorption processes. Further, the maximum adsorption capacities of the synthesized PUL-NFs increased with increasing the amount of CNDs embedded in these mats. The adsorbent M4 performed better by removing the highest amount of STZ from aqueous media, depicting its potential for remediating wastewater streams polluted with STZ.

### 3.3. STZ Removal Mechanism

The affinity of an adsorbate towards a specific adsorbent is mainly related to the features of both adsorbate and adsorbent as well as adsorption conditions. Thus, the mechanism of a contaminant removal varies with the type of the adsorbate and adsorbent. For instance, the adsorption of ionic compounds is generally controlled by electrostatic interactions between the adsorbate and the adsorbent. As STZ molecules generally exist in ionic forms in aqueous media, therefore, electrostatic interactions may play a role in its removal from water. However, the speciation of the STZ molecule depends on the solution pH. The STZ^+^ exists in strong acidic pH (~3.0) and STZ^-^ exists in neutral to slightly alkaline pH (~7.4), and STZ^0^ exists between these pH ranges [33], suggesting the potential of electrostatic interactions for governing the STZ removal mechanism. A higher STZ adsorption onto soil particles has earlier been reported in acidic conditions due to electrostatic interactions between STZ^+^ and negatively charged soil particles [37]. In contrast, in the current study, based on the pH of the solution, STZ^0^ and STZ^–^ were the dominant species. Nonetheless, the highest STZ removal was seen in the pH range of 4.0–6.0 in all the adsorbents, suggesting that electrostatic interactions were not the dominant removal mechanism, owing to the predominance of STZ^0^ ions (Figure 7). However, a sharp decline in STZ adsorption after pH 7.0 could be due to electrostatic repulsions between STZ^–^ and the –OH^-^ groups on PUL-NFs [38]. On the other hand, the highest adsorption of STZ in the pH range of 4.0–6.0 could be owing to Lewis acid–base effects and electron donor–acceptor (EDA) π-π connections [39]. The presence of sulfonamide and amino groups on the aromatic structure of STZ might have enabled it to interact with –OH groups present on the surface of PUL-NFs, consequently removing the STZ^0^ in the acidic pH range by Lewis acid–base reactions.

As CNDs can act both as electron acceptor and donor, stronger π-π EDA interactions could be the reason for higher STZ removal in the acidic pH range [34]. Due to the presence of N-heteroaromatic rings and a lone pair of electrons, STZ molecules have the potential to accept π-electrons under acidic pH, whereas the functional groups of PUL-NFs can serve as electron donor–acceptor sites [40]. Therefore, stronger EDA π-π interactions were developed by the transfer of electrons from the –OH groups of PUL-NFs to the amino group of STZ. Additionally, the –OH groups of PUL-NFs developed H-bonding with the –NH groups of the STZ at higher pH levels and aided the STZ removal process [41]. Furthermore, STZ removal was further aided by the diffusion of adsorbate molecules into the porous structure of the adsorbent, as indicated by the intraparticle diffusion model [42]. Therefore, owing to highly porous structures and large surface area, the addition of CNDs increased the diffusion process and subsequently increased STZ removal [40]. Moreover, simulations of kinetics and isotherm models suggested single and multi-layer chemical adsorption of STZ onto the synthesized PUL-NFs.

Post-adsorption XRD and ATR-FTIR analyses were used to confirm the adsorption of STZ onto PUL-NFs adsorbents. A new peak was evidenced at 23.1° in post-adsorption XRD patterns that was designated as STZ (Figure 4b). The peak intensities of all graphene-like carbons reduced after the adsorption of STZ, suggesting the successful loading of STZ molecules on PUL-NFs. Contrarily, the intensity of the peak at 34.8° (graphene quantum dots) was increased with STZ adsorption. Likewise, the observed peak of N nearby 35.8° disappeared after adsorption of STZ. Similarly, the variation in band intensities and the appearance of new bands in FTIR spectra depicted the adsorption of STZ onto PUL-NFs (Figure 5b). The intensities of the bands around 1020 and 2850 cm^−1^ were increased after STZ adsorption. Therefore, EDA π-π interactions and Lewis acid–base reactions were the dominant mechanisms for STZ removal under acidic pH levels, which were further aided by STZ diffusion into the pores, whereas H-bonding and diffusion resulted in STZ removal from aqueous solution at higher pH levels.

To achieve the aim of this study, which was designed to keep the environment clean by fabricating pullulan-based nanofibers with CNDs to remove STZ, the adsorbents after the adsorption study were collected successfully from the aqueous medium. Unfortunately, all the collected membrane can only be used one time because all the membranes become hard after the adsorption study. The digitals images of M1, M2, M3 and M4 are shown below (Figure 10).

## 4. Conclusions

In this work, CNDs-embedded PUL-NFs membranes were fabricated by the electrospinning technique and successfully employed for STZ removal from water. PUL was loaded with different concentrations of CNDs, ranging from 0.0% to 3.0%, and characterized. It was found that the maximum adsorption capacity occurred in the pH range of 4.0 to 6.0. Furthermore, increasing the amount of embedded CNDs in PUL resulted in an increase in the STZ adsorption capacity, due to the distinctive properties of the nanomaterials and the overall surface area of the prepared nanofiber membranes. Simulations of kinetics and isotherm models suggested single and multi-layer chemical adsorption of STZ onto synthesized nanofiber membranes. Overall, the addition of 3.0% of CNDs in PUL-NFs removed the maximum amount of STZ (133.68 mg·g^−1^) from aqueous solutions. Furthermore, FTIR and XRD analyses after STZ adsorption confirmed the presence of STZ on the surface of nanofiber membranes. Lewis acid–base reactions and π-π electron donor–acceptor interactions were the dominant mechanisms, along with H-bonding and diffusion, for STZ removal. Thus, the synthesized CNDs-embedded PUL-NFs membranes exhibited excellent physical and chemical properties, making them efficient adsorbents for STZ removal from wastewater.

## Data Availability

Not applicable.

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
