# Peer review of "Carbon Nanodots-Embedded Pullulan Nanofibers for Sulfathiazole Removal from Wastewater Streams"

_membranes, 2022, doi:10.3390/membranes12020228_

Round 1
Reviewer 1 Report
In this work, the authors used pullulan (PUL) nanofibers doped with carbon particles for sulfathiazole (STZ) removal. There is some improvement of the STZ adsorption after adding carbon particles, with the pH effect and adsorption behavior further investigated in the paper to understand the adsorption mechanism. Yet, there remain some aspects to improve:
- Figure 2 and line 234-235 at page 8, to identify the carbon particles to be carbon quantum dots (CQDs), more experimental evidence is needed. CQDs are referred to small carbon nanoparticles below 10 nm. ( Mater. Chem. C, 2014,2, 6921-6939.) However, Figure 2 shows that the carbon particles are much larger in the study.
- Figure 3, the loading of carbon particles on PUL is very small (< 4% w/v ratio), and therefore SEM of carbon particles before loading onto PUL may better present the morphology of carbon particles.
- Since the work specifically studies the STZ removal from wastewater stream, a more complete introduction should be included, such as the state-of-the-art approach, relevant STZ concentration, etc. Evaluation of the efficiency of this work compared with other works can also be included.
- Page 13 line 353, the improved STZ adsorption is attributed to the enhanced surface area after adding carbon particles, which lacks experimental support such as BET-specific surface area.
- Page 13 line 367, “M3 and M2” seems like a typo, it should be “M3 and M4”. Page 14 line 395, “M2 (38.79)” should be “M1(38.79)”.
- Page 17 lines 470-472, the earlier report on soil particles lacks a reference.
- M4 sample, which has the highest loading of carbon particles, shows the best adsorption performance. Will it be further enhanced by adding more carbon particles? What about pure carbon particles for STZ removal?
- Can the materials be reused or recycled? How stable the performance is?
Author Response
Thanks for your comments. Our responses are uploaded in the attached file.

Reviewer 2 Report
Review comments:
The manuscript “carbon quantum dots-embedded pullulan nanofibers for sulfathiazole removal from wastewater streams” mainly reported a new nanofiber for sulfathiazole removal. The produced nanofiber showed high adsorption capacity (133 mg g-1), and the adsorption mechanism is Lewis acid-base reaction and π-π electron donor-acceptor interactions. This reported new nanofiber was fabricated from leaves, wastage from date palm trees, and it is an environmentally friendly raw material. I think this paper can be accepted if author could solve following drawbacks.
- In abstract, M4 with high concentration of CQDs (3.0%, w/w) showed the superior adsorption performance compared to other samples with low CQDs loading. How about the higher concentration of CQDs (> 3.0%, w/w)?
- Page 3, “However, CQDs-embedded NFs have not yet been used for the adsorption of STZ antibiotics from water”. c.
- In figure 2, the marked CQDs is very large, and it is nor consistent with the result of TEM in Figure 3. Please provide more evidences.
- Page 11, line 309, “TGA and DTA observations are presented in Table a, indicating that as percentage of CQDs increased, the onset temperature and maximum peak degradation temperature were increased along with residue percentage due to higher amounts of CQDs”. However, as shown in Table 1, M4 with the highest concentration of CQDs showed the smallest onset temperature than that of M1.
- “The XRD patterns of all the materials exhibited two very intense peaks at 44.5º and 78.1º, depicting the presence of CQDs.” However, M1 without CQDs also showed the characteristic peaks of CQDs.
- “The rapid adsorption phase in M3 and M4 during the initial 10 min could be due to the abundance of the sorption sites, ….” Did you check the change in hydrophilicity/hydrophobicity of PUL after embedding CQDs? I think that in addition to the increase in adsorption sites, the altered hydrophilicity/hydrophobicity of PUL-NFs (M3, M4) may also be responsible for the accelerated adsorption of STZ at the initial stage (liquid film diffusion).
- Adsorption isotherms. The isotherm data are insufficient and inadequate to support the fit of several isotherm models. It is suggested to add more data points appropriately.
- The repeated stability and regeneration strategy should be considered.
Author Response

(The authors gave the same response as above.)

Reviewer 3 Report
In this work, CQDs-embedded pullulan nanofibers were developed and applied for sulfathiazole removal from wastewater streams. Although this work has achieved some valuable results, there are some issues that should be clarified and more experiments and discussion should be supplied before further consideration.
- Figure 3 showed the morphology of CQDs embedded in pullulan nanofibers. However, the TEM images showed the obvious aggregation of CQDs. Did the aggregation of CQDs influence the removal efficiency of sulfathiazole? Can the dispersion of CQDs be improved? On the other hand, the size of the as-prepared CQDs should be supplied.
- Raman spectrum is required to be carried out for characterizing the degree of graphitization of CQDs.
- Can the authors provide the mechanical properties (e.g. tensile tests) of CQDs-embedded pullulan nanofiber?
- A schematic diagram should be supplied for illustrating the STZ removal mechanism.
Author Response

(The authors gave the same response as above.)

Round 2
Reviewer 1 Report
Most questions in the review have been clearly answered and addressed.
However, it might be misleading to use “carbon quantum dots (CQDs)” to describe the carbon nanoparticles since CQDs are small carbon nanoparticles below 10 nm. (J. Mater. Chem. C, 2014,2, 6921-6939.) According to the author’s previous work describing the carbon particles preparation ( www.nature.com/articles/s41598-020-73097-x. ), the carbon nanoparticles are “carbon nanodots”.
Reviewer 3 Report
The revised manuscript can be accepted for publication.
